# Prevalence, Radiographic Features and Clinical Relevancy of Accessory Canals of the Canalis Sinuosus in Cypriot Population: A Retrospective Cone-Beam Computed Tomography (CBCT) Study

**DOI:** 10.3390/medicina58070930

**Published:** 2022-07-14

**Authors:** Zafer Beyzade, Hasan Güney Yılmaz, Gürkan Ünsal, Ayşe Çaygür-Yoran

**Affiliations:** 1Department of Periodontology, Faculty of Dentistry, Near East University, Mersin 99138, Turkey; guney.yilmaz@neu.edu.tr (H.G.Y.); ayse.caygur@neu.edu.tr (A.Ç.-Y.); 2Department of Dentomaxillofacial Radiology, Faculty of Dentistry, Near East University, Mersin 99138, Turkey; gurkan.unsal@neu.edu.tr

**Keywords:** canalis sinuosus, cone-beam computed tomography, accessory canal

## Abstract

*Background and Objectives:* This retrospective study aims to evaluate the prevalence, radiographic features, and clinical relevancy of the accessory canals (AC) of the canalis sinuosus (CS) in patients referred for implant surgery. *Materials and Methods:* Cone-beam computed tomography (CBCT) images of the patients were collected and ACs were evaluated. Age, sex, bilateral distribution, localization, diameter, distance to the buccal cortical plate, distance to the crest of the alveolar ridge, terminal ending localization, and the presence of tooth or implant were recorded. Ninety-one patients who were eligible for this study were enrolled. *Results*: A total of 188 ACs were found in 91 patients with 86 bilateral and 5 unilateral distributions. The mean age of the patients was 45.39. All ACs had a terminal ending at the palatal cortical border. All parameters showed a non-normal distribution; thus, the Mann–Whitney U test was preferred. Bilateral AC distribution (*p* = 0.761), AC distance to the crest of the alveolar ridge (*p* = 0.614), AC distance to the buccal cortical plate (*p* = 0.105), and AC diameter (*p* = 0.423) showed no significant difference between females and males. According to our study, a CS might be an anatomical structure rather than an anatomical variation, as all patients had at least one AC of the CS. It can be inferred that the detection of ACs will be achievable once clinicians are aware of these structures with continuous regular anatomy reworks and with small voxel-sized CBCT devices. *Conclusion:* This study was conducted to find the features and prevalence of the CS, and it was found that the CS is an anatomical structure rather than an anatomical variation. This argument is in line with the information on the CS in Gray’s Anatomy, 42nd Edition. Impaired healings and complications of the CS can be avoided if clinicians follow the American Academy of Oral and Maxillofacial Radiology guidelines regarding pre-operative implant examination. Otherwise, avertible complications may cause significant impairments in quality of life.

## 1. Introduction

The tracts of the anterior superior alveolar nerve (ASAN) and its intraosseous canal, the canalis sinuosus (CS), were described in “The Anterior Superior Alveolar Nerve and Vessels” by Frederic Wood Jones in 1939 [1]. The CS innervates the maxillary sinuses, the anterior teeth, and the nasal fossa floor. Several synonyms are used in the literature, such as lateral incisor canal, sinuous canaliculus, anterior superior alveolar canal, or neurovascular variation in the anterior palate [2,3,4,5,6,7,8]. It is named for its double-curved course and is accepted as a major branch of the infraorbital nerve, since it runs for 55 mm through the maxilla [8,9].

The CS advances in an anterior-lateral path from the canalis infraorbitalis to the foramen infraorbitalis. As it arrives at the maxilla’s anterior wall, the course of the CS alters and takes a medial turn to proceed inferior to the infraorbital foramen, medially crossing the maxillary sinus wall. As it reaches the apertura piriformis, it deviates inferiorly along and reaches the maxillary anterior region. The CS involves the associated arterial and venous blood vessels, in addition to the ASAN. The first remarkable branch is generally found in the canine tooth area and other accessory canals (AC) are found around the maxillary incisor region [1,8,10,11]. Several studies have been performed on the prevalence of the CS in specific populations. Ghandourah et al. [12] reported the prevalence as 65.75% in 219 German patients, and Tomrukçu et al. [13] reported a 34.7% prevalence of CS in Turkish patients. Santos et al. [14] reported a 52.1% to 88% prevalence of CS and concluded that some articles suggest that instead of being an anatomical variation, the CS can be an anatomical structure.

As it is reported that the CS may have multiple variations in the maxillary anterior region, conventional 2D radiographs are not sufficient to detect them. Even those variations may cause peri-operative or post-operative complications, as they are rarely identified by clinicians, and they can also be misdiagnosed as apical osteitis [8,10,13]. Routine radiographic examinations are often unsuccessful in the imaging of ACs due to radiological limitations, e.g., low image quality, magnifications, superimpositions, and distortions. Even if CSs are visible on the 2D radiographs, most practitioners are unaware of their presence; thus, those structures may be confused with apical pathologies, since they are seen as the periapical radiolucencies in the region of the upper incisor. Cone-beam computed tomography (CBCT) plays an essential role in anatomical variation detection due to its ability to produce high-resolution 3D cross-sectional images with lower costs and at a lower radiation dose. Thanks to CBCT, the neurovascular bone channels of the maxilla can be visualized in finer detail. According to the literature, the best way to interpret CSs is through CBCT examinations. CBCT slices demonstrate a twisted bony channel that originates from the infraorbital canal and extends to the anterior maxilla [4,10,11,12,13,15,16,17,18].

The purpose of this retrospective study is to evaluate the prevalence, radiographic features, and clinical relevancy of ACs of the CS in implant procedures in the Cypriot population with pre-acquired CBCT images, as the prevalence of the CS differs in different populations and in different studies.

## 2. Materials and Methods

### 2.1. Ethical Considerations

In this retrospective study, CBCT data collection and evaluation were carried out at Near East University, Faculty of Dentistry, Department of Dentomaxillofacial Radiology. The study was conducted by the Near East University Ethics Review Board, Health Sciences Ethics Committee, with the approval of YDU/2021/91-1350. The study was performed according to the declaration of Helsinki.

### 2.2. EQUATOR Guideline

This study is in accordance with the EQUATOR Network’s CRIS Guidelines. This study meets the criteria mentioned: “sample size calculation, meaningful difference between groups, sample preparation and handling, allocation sequence, randomization, and blinding statistical analysis”.

### 2.3. Data Collection

This retrospective study was designed to analyze CBCT images that were pre-taken for various dental complaints (such as apical lesion, impacted tooth, dental trauma, implant treatment, dental anomalies, temporomandibular joint disorders, periodontal bone defects) at Near East University, Faculty of Dentistry, Department of Dentomaxillofacial Radiology, between 2019 and 2021. CBCT images were taken in high-quality mode with the Sirona Orthophos SL 3D CBCT Unit (Dentsply Sirona, Bensheim, Germany) in 2 Field of View (FOV) options. The FOVs used for this study were 80 mm × 80 mm and 110 mm × 100 mm. The imaging parameters ranged between 60 and 90 kVp and 3 and 16 mA, depending on the FOV. The voxel size was 0.08 mm, and the scanning time was 14 s. The 2D measurements were carried out with the Sidexis 4 Imaging Software (Dentsply Sirona, Bensheim, Germany) with Advantech KT R240FEE Medical LCD Monitor (Advantech by Kostec, Gangwon, South Korea) in order to achieve a higher-quality radiographic examination. Since this study examines the ACs of CS in the Cypriot population, some inclusion and exclusion criteria were defined.

Inclusion criteria were (Table 1):Patients of Cypriot origin;Fields of View which demonstrated all maxilla (from the apertura piriformis to the alveolar crest).

Exclusion criteria were (Table 1):
Images with motion artefacts;Images with beam-hardening artefacts;Images with intraosseous pathologies (cysts, tumors) which are localized in the anterior maxilla;Patients with cleft palate;Patients with syndromes that affect dentomaxillofacial structures;Patients with an operation history in the anterior maxilla;Patients with maxillary fractures.

The parameters for the patients with AC were (Table 1):
Age;Sex;Presence of AC;Number of ACs;Unilateral/bilateral distribution;Localization #1 (FDA Dental Notation System was used for patients with teeth);Localization #2 (axial distance to nasopalatine duct was measured for edentulous patients);Diameter of the ACs;AC distance to buccal cortical plate;AC distance to alveolar crest;Palatal or buccal localization of the AC opening.

### 2.4. Radiological Examination

The examination started with identifying the intraosseous canals in the maxillary anterior region with an upward tract to the CS. Bilateral/unilateral distribution was then noted with axial, coronal, and sagittal slices. Age, sex, and the number of intraosseous canals were registered. The exact measurement was noted if the AC’s diameter was wider than 1 mm. The canals with a diameter of less than 1 mm were noted as “1>”. Diameters were determined from cross-sectional images by measuring the opening at the palatal cortical plate. As this study investigates the prevalence of ACs in specific tooth areas for implant procedures, the modified classification of Oliveira-Santos et al. [17] was used. The maxillary anterior region was divided into 7 zones: the central incisor zone, between the central and lateral incisor, the lateral incisor zone, between the canine and lateral incisor, the canine zone, between the canine and the first premolar, and the first premolar zone. The diameter of the terminal ACs was measured in cross-sectional and sagittal CBCT slices. Distance to the buccal cortical plate was measured in cross-sectional CBCT images, drawing a linear line from the anterior border of the terminal ending of AC to the buccal cortical plate. Distance to the alveolar crest was measured in cross-sectional CBCT images by drawing a vertical linear line from the AC’s opening to the alveolar ridge crest. Palatal or buccal opening of the ACs was also recorded in the sagittal slices. The study data also noted the presence of a tooth or implant.

### 2.5. Statistical Analysis

Three clinicians, H.G.Y., G.Ü., and Z.B., a periodontology professor with 15 years of experience with CBCT, a dentomaxillofacial radiologist, and a periodontology Ph.D. student, respectively, were calibrated for all measurements using CBCT software programs. All authors discussed the protocol by drawing schematic diagrams and agreed on the proposed method for evaluating the associated data. The final decisions for all measurements were made during a consensus meeting of the 3 investigators in the case of a different categorical result. Those disagreements were resolved by the dentomaxillofacial radiologist G.Ü., and G.Ü. re-evaluated all images after a 2-week interval. The first phase of the study included the detection of the ACs in multi-planar slices. The second phase of the study included examining the ACs according to the above-mentioned parameters.

A descriptive statistical analysis of the results was performed. An independent samples T-test was used to assess the normally distributed parameters, while the independent samples nonparametric test was used to assess the non-normally distributed parameters between genders. Statistical significance was set at the level of *p* < 0.05. The IBM SPSS Statistics (IBM Corp., Armonk, NY, USA) version 24.0 software was used for statistical analysis.

## 3. Results

One hundred and nine Cypriot patients with CBCT images of the maxillary anterior region were found. According to our exclusion criteria, 18 cases were excluded. The cases were:Twelve cases with motion or beam-hardening artifacts;A case with an AC which was in contact with a radicular cyst (Figure 1);A case with a cleft palate operation;A case with a surgically assisted rapid palatal expansion (SARPE) operation;A case with an odontogenic tumor (compound odontoma);A case with a hemifacial microsomia;A case with a cleft palate.

A total of 91 Cypriot patients (52 males, 39 females) eligible for this study were enrolled. The mean age of the patients was 45.39 (min 11, max 74). In total, 188 ACs were found in 91 patients with 86 bilateral and 5 unilateral distributions.

Localization was evaluated in seven zones: the central incisor zone, between the central and lateral incisors, the lateral incisor zone, between the lateral incisor and the canine, the canine zone, between the canine and the first premolar zone, and the first premolar zone. The number of cases was 49, 28, 88, 11, 7, 3, and 2 in each zone, respectively. The majority of the ACs, 169 canals, had diameters smaller than 1 mm, and 19 canals had diameters bigger than 1 mm (max 2.79 mm).

The mean distance to the buccal cortical plate was 7.35 mm (min 3.0 mm, max 12.08 mm), and the mean distance to the crest of the alveolar ridge was 5.87 mm (min 0 mm, max 17.03 mm). In total, 17 canals were exposed at the crest of the alveolar ridge (Figure 2), 21 canals were between 0.1 and 2.99 mm away from the crest, 51 canals were between 3 and 4.99 mm away from the crest, 51 canals were between5 and 7.99 mm away from the crest, and 48 canals had at least 8 mm distance from the crest (Figure 3).

All ACs had a terminal ending at the palatal cortical border, while no canal was found to end at the buccal cortical plate (Figure 4). All parameters showed a non-normal distribution between the genders; thus, according to the results of the Mann–Whitney U test, bilateral AC distribution (*p* = 0.761), AC distance to the crest of the alveolar ridge (*p* = 0.614), AC distance to the buccal cortical plate (*p* = 0.105), and AC diameter (*p* = 0.423) had no significant difference (Table 2).

## 4. Discussion

The CS prevalence was reported to be between 15.7% and 100% by Rusu et al. [19], and Aoki et al. [20] stated that 54.14% of the patients had bilateral CS presence; however, in this study, 100% of the cases had ACs and 95.60% of the patients had bilateral canals, which is the highest percentage recorded in the literature. There are multiple factors which can lead to this result, as the imaging modality, voxel sizes, inclusion and exclusion criteria of the study, observer’s experience, and presence/absence of artefacts can affect the number of cases [4,10,11,12,13,15,16,17,18,21,22]. Since there is no uniform method for evaluating CSs and ACs, all articles should be evaluated according to their methods section. By way of example, the majority of the CT devices have anisotropic voxels, which rule out a multi-planar evaluation that can alter the number of ACs. Voxel sizes differ between CBCT units, and as there are ACs whose diameter is smaller than 1 mm, any device with a voxel size bigger than 1 mm may be unable to demonstrate them. Some articles excluded ACs with a diameter smaller than 1 mm, thus, this criterion directly affects the evaluation. The detection of ACs requires a perfect differential diagnosis and a high awareness, since artefacts and wide trabeculation areas may be misinterpreted as AC. Three trained investigators conducted this study to avoid any error related to the mentioned criterion. The presence of artefacts, especially the beam-hardening artefact, may make it impossible to detect some ACs; therefore, we excluded all images with an artefact between the maxillary first premolar areas.

Misdiagnosis-related injuries, either due to the clinician’s lack of knowledge or to the imaging modality, can cause noteworthy complications, especially in implant procedures. Other procedures in which the CS can become damaged are reported as horizontally impacted canines, antrostomy, Le Fort I osteotomies, and fossa Canina punctures. Midface pain associated with ASAN injury is reported to be felt in the maxillary anterior region, and 16 out of 25 Le Fort I osteotomy patients have been reported to have somatosensory disturbances [3,5,21,23,24]. As the opening of the ACs of the CS is generally localized on the palatal cortical plate in the maxillary anterior region, clinicians should search thoroughly for CS in CBCT slices [5,10,13,18,20,23,24,25,26,27,28]. Multiple authors have reported dental implant placement-related hemorrhage and neurosensory damage cases. The common findings are paresthesia, pain, localized infection, and failed osseointegration [5,23,26,28,29].

Lopes Dos Santos et al. [14] evaluated seven potential CS injuries and stated that five of them involved implant treatments. They reported that four out of five patients felt intense pain and discomfort only a few hours following the surgery and that one patient even had nasal bleeding with a swollen area in the subnasal region. They also reported that exposed CSs might cause stabbing and sharp pain, which is not associated with the stimulus intensity in the presence of mechanical stimulation. Wanzeler et al. [11] also reported such exposed CSs and stated that this situation results in infeasible sites for implant treatment. Seventeen patients with exposed AC were found in this study, and none described such a pain in their anamnesis form.

Volberg et al. [28] reported an extraordinary case with intense pain and paresthesia at the implant site, which increased after the first post-operative week and was not relieved by the analgesics. A burning pain that was felt in the occipital region was also present around the left maxillary canine, the palate, and the nasal region. They found that the only nerve block which relieved the pain was the infraorbital block. The second evaluation found a CS with a 2.3 mm diameter in the pre-operative CBCT images. Following the extraction of the implant, the paresthesia and pain faded; however, a necrotic area appeared on the palatal mucosa with a compromised palatal blood supply. The authors stated that “the contraction of smooth muscle within the arterial wall during the neurovascular bundle constriction” may lead to necrosis.

Machado et al. [7], McCrea et al. [29], and Arruda et al. [5] presented dental implant cases with persistent pain in the maxillary anterior region that were relieved following the implant extraction. However, Olenczak et al. [24] reported another case with the same onset but uneventful healing after the extraction. The patient in the latter report did not have any relief even after the extraction of the implant at the maxillary left first premolar area. Shintaku et al. [23] also reported three cases of neurovascular damage of the CS and secondary implant failure. Hong et al. [30] and Kalpidis et al. [31] underlined that even the CSs and ACs with diameters smaller than 1 mm might cause hemorrhage following the implant placement. In light of these publications, ACs with diameters smaller than 1 mm were also evaluated in this study, and it was found that those ACs constitute 89.89% (169 of 188) of the ACs in this study.

Although some patients described post-operative pain due to psychological issues, Balaji states that clinicians should evaluate patients by means of CS and AC in the presence of post-operative pain and bleeding [29]. The clinicians should evaluate the maxillary anterior region with small voxel-sized CBCT slices to maintain a thorough differential diagnosis, since neglecting the presence of these canals might delay the neurovascular damage diagnosis. Ferlin et al. [16] stated that the standardization of the voxel sizes is also essential, as they are crucial in detecting thin and small anatomical structures. Their study was carried out with voxel sizes of 0.3 mm. The current study was conducted with 0.08 mm voxel sizes, allowing us to evaluate most of the ACs in multiple slices.

According to the results of the present study, the mean distance to the buccal cortical plate was 7.35 mm (min 3.0 mm, max 12.08 mm), and the mean distance to the crest of the alveolar ridge was 5.87 mm (min 0 mm, max 17.03 mm). Since the difference between the min and max values of the distance to the buccal cortical plate is 9.08 mm and the difference between the min and max values of the distance to the crest of the alveolar ridge is 17.03 mm, it is crucial to state that presumptive evaluations must be avoided.

Machado et al. [7] reported that only 5.1% of ACs had a terminal ending at the buccal cortical plate, and Tomrukçu et al. [13] reported that 3 out of 214 ACs had a terminal ending at the buccal cortical plate. The significant finding for the Cypriot population was that all ACs terminal endings were located on the palatal cortical plate.

Yeap et al. [32] also conducted a study to examine the CS in the Australian population using CBCT, and they reported at least one CS on 198 scans with 98.5% prevalence. The high prevalence reported in their study is in accordance with our results. They also stated that their lowest prevalence was seen with the 0.25 mm voxel size; their voxel sizes varied between 0.08 mm and 0.25 mm. Thus, it can be concluded that small voxel size plays a crucial role in the detection of the ACs of CS. Moreover, they reported that only 4 out of 412 ACs were located at the labial cortical plate, while 408 ACs, 99%, were located at the palatal cortical plate, which is also in accordance with our results. Fernandes et al. [33] conducted a study to find the prevalence of the canalis sinuosus on the alveolar ridge at the site of endo-osseous implant placement in the Chennai population, and reported that the prevalence of ACs amongst male patients was 14% and that it was 4% for female patients. They also stated that the accessory canals of the canalis sinuous in the site of endo-osseous implant placement are rare among their participants. Shan et al. [15] examined the CBCT images of 1007 patients to find its prevalence in the Chinese population and reported that 36.9% of the CBCT scans demonstrated at least one AC of the CS. They also stated that all ACs perforated the palatal cortical plate, as we also reported in this study. However, given that their voxel size was 0.2 mm, two and a half times the size of our voxels, it is logical to state that they could not to identify smaller ACs.

It can be inferred that the detection of ACs will be achievable once clinicians are aware of these structures with continuous regular anatomy reworks and with small voxel-sized CBCT devices. This study was conducted to find the features and prevalence of the CS, and it was found that the CS is an anatomical structure rather than an anatomical variation. This argument is in line with the information on CS in Gray’s Anatomy, 42nd Edition [34]. Impaired healings and complications of the CS injuries can be avoided when clinicians follow the American Academy of Oral and Maxillofacial Radiology guidelines regarding the pre-operative implant examination [35]. Otherwise, avertible complications may cause significant impairments in quality of life. This study had a limitation. A larger sample size might have revealed more features of ACs; however, in order to limit the adverse effects of the pathologies and artefacts, we had some relatively broad exclusion criteria; thus, only 91 Cypriot patients’ CBCT scans were evaluated.

## 5. Conclusions

The maxillary anterior region is generally considered to be a relatively complication-free site. However, due to the presence of ACs of the CS, pre-operative implant examinations should be carried out thoroughly with small-voxel sized CBCT devices, as the American Academy of Oral and Maxillofacial Radiology recommends, in order to prevent complications such as discomfort, mid-face pain, paresthesia, post-op hemorrhage, nasal hemorrhage, localized infection, subnasal swelling, failed osseointegration, and necrosis.

## Figures and Tables

**Figure 1 medicina-58-00930-f001:**
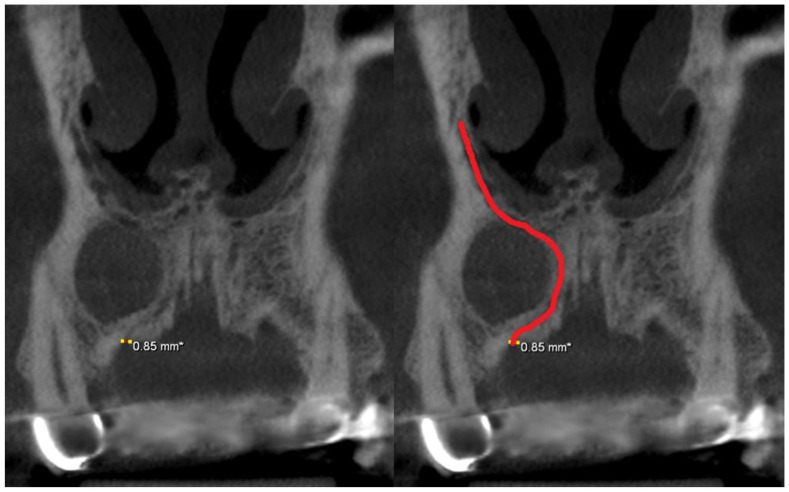
Coronal CBCT slices of an excluded case of an AC (**red line**) which was displaced due to a radicular cyst. Note that this accessory canal with a 0.85 mm diameter, ends at the crest of the alveolar ridge.

**Figure 2 medicina-58-00930-f002:**
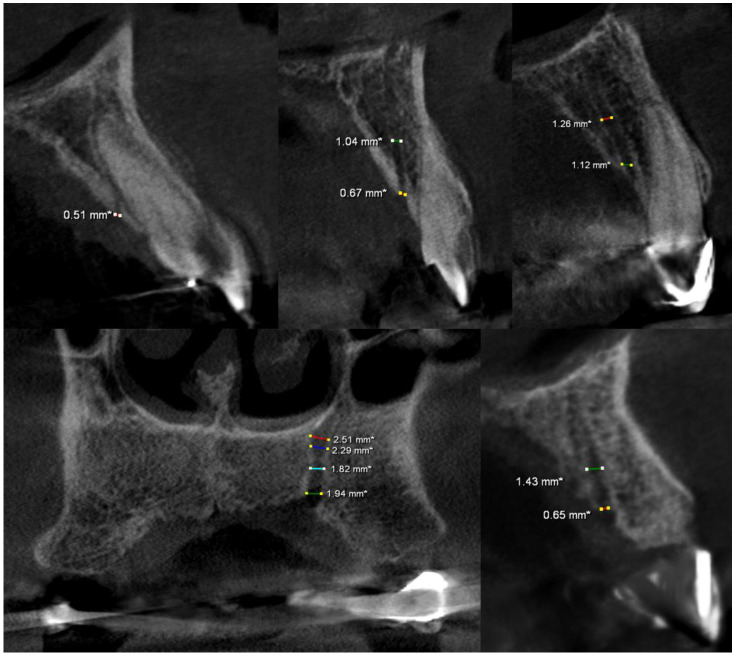
Sagittal and coronal CBCT slices of various exposed ACs cases. Note that the AC curve, extension, and diameter may vary in each case. Red-Blue-Cyan-Green lines demonstrate different diameters at different levels.

**Figure 3 medicina-58-00930-f003:**
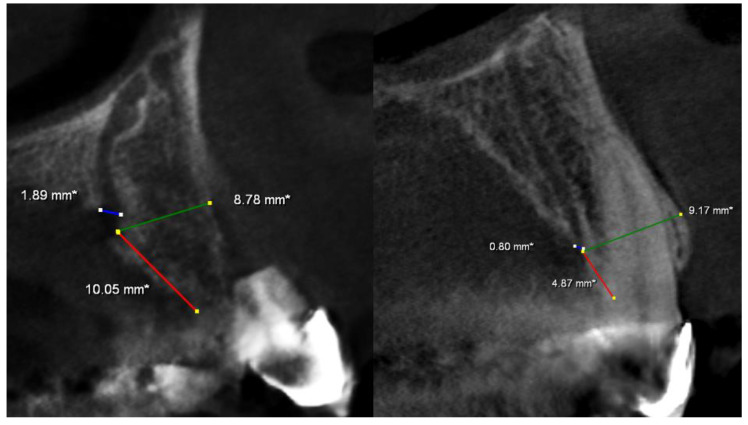
Radiological examination pattern of the AC diameter (**blue line**), AC distance to the buccal cortical plate (**green line**), and AC distance to the crest of the alveolar ridge (**red line**).

**Figure 4 medicina-58-00930-f004:**
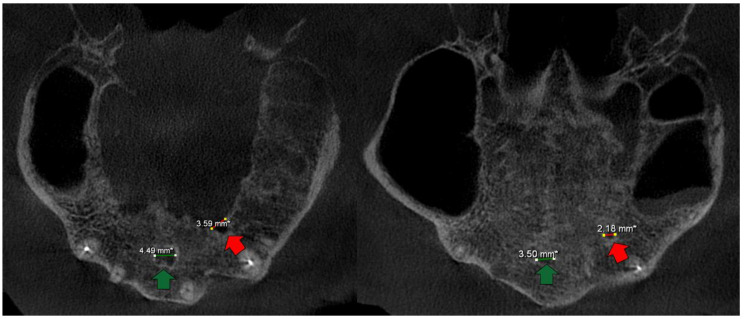
Axial CBCT slices from a patient with an exposed AC which ends at the palatal cortical border. The green arrow shows the nasopalatine canal, while the red arrow shows the AC. Note that in some slices an AC as wide as the nasopalatine canal can be seen.

**Table 1 medicina-58-00930-t001:** Summary of the materials and parameters of the study.

CBCT Device	Orthophos SL 3D CBCT Unit (Dentsply Sirona, Bensheim, Germany)
Field of view selection (mm × mm)	80 × 80/110 × 100
Imaging parameters	
Peak potential (kVp)	60–90
Currency (mA)	3–16
Voxel size (mm)	0.08
Scanning time (seconds)	14
Imaging software	Sidexis 4 Imaging Software (Dentsply Sirona, Bensheim, Germany)
Medical monitor	Advantech KT R240FEE Medical LCD Monitor (Advantech by Kostec, Gangwon, South Korea)
Inclusion criteria	Patients of Cypriot origin
	Images that cover from the apertura piriformis to the alveolar crest of the maxilla
Exclusion criteria	Images with motion artefacts
	Images with beam-hardening artefacts
	Images with intraosseous pathologies (cysts, tumors) which are localized in the anterior maxilla
	Patients with cleft palate
	Patients with syndromes that affect dentomaxillofacial structures.
	Patients with an operation history in the anterior maxilla
	Patients with maxillary fractures
Study parameters	Age
	Sex
	Presence of AC
	Number of ACs
	Unilateral/bilateral distribution
	Localization #1 (FDA Dental Notation System was used for patients with teeth)
	Localization #2 (axial distance to nasopalatine duct was measured for edentulous patients)
	Diameter of the ACs
	AC distance to buccal cortical plate
	AC distance to alveolar crest
	Palatal or buccal localization of the AC opening

**Table 2 medicina-58-00930-t002:** Statistical analysis of the parameters among females and males.

Parameters	*p* Value
AC Diameter—Gender	0.423
AC Distance to the Buccal Cortical Plate—Gender	0.105
AC Distance to the Crest of the Alveolar Ridge—Gender	0.614
AC Distribution—Gender	0.761

## Data Availability

The data sets used and analyzed during the current study are available from the corresponding author on reasonable request. The data are not publicly available due to privacy/ethical restrictions.

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
