# Peer review of "Prevalence, Radiographic Features and Clinical Relevancy of Accessory Canals of the Canalis Sinuosus in Cypriot Population: A Retrospective Cone-Beam Computed Tomography (CBCT) Study"

_medicina, 2022, doi:10.3390/medicina58070930_

Round 1

Reviewer 1 Report

Abstract: Conclusion missing

Introduction: What is the reseach gap? Why authors want to conduct and puvblished this paper?

Method: Put information from line 97 to 121 into a Table.

Results: Figure 1 should be a typical radiograh of the rults and should be in result section. Table I should present data and delete the test column (no need to mention MWU test)

Discussion:

The aim of the study is to evaluate the incidence, radiographic features, and clinical significance of accessory canals of the canalis sinuosus in patients referred for implant surgery. Similar papers were published as

1 Examination of canalis sinuosus using cone beam computed tomography in an Australian population. Aust Dent J. 2022 Mar 14. doi: 10.1111/adj.12910.

2 CBCT Analysis of Prevalence of the Canalis Sinuosus on the Alveolar Ridge in the Site of Endosseous Implant Placement: A Retrospective Study. J Long Term Eff Med Implants. 2022;32(2):45-50. doi: 10.1615/JLongTermEffMedImplants.2022039656.

3. Cone beam computed tomography analysis of accessory canals of the canalis sinuosus: A prevalent but often overlooked anatomical variation in the anterior maxilla. J Prosthet Dent. 2021 Oct;126(4):560-568. doi: 10.1016/j.prosdent.2020.05.028. Epub 2020 Sep 29.

The author should discussed their finding with references to all these papers.

This is a retrospective study and thus its limitations must be discussed in details.

The conclusion is too long. Move the information to discussion. Please trim it to address the objective of the study.

Author Response

1-) We added the proper conclusion to the abstract as the reviewer#1 suggested.

2-) We agree with the reviewer#1, at the last sentence of the introduction section we clarified our real goal and the reason behind it as the reviewer#1 suggested.

3-) We would like to thank the reviewer#1 with this suggestion. We added a new table and we also believe that this table will be more reader-friendly.

4-) List and placement of figures are revised as the reviewer #1 suggested. MWU test is removed from the Table as well. 

5-) As the reviewer#1 suggested, we discussed the recommended articles' findings with references.

6-) We discussed our major limitation with details in the final paragraph of the discussion section as the reiewer#1 suggested.

7-) We agree with the reviewer#1. We added the first half of the conclusion to the discussion with rewritting that part.

8-) As reviewers suggested we revised our manuscript and corrected some typos.

Reviewer 2 Report

How could a retrospective study estimate incidence?

What was the indication for cbct imagine?

It would be better to replace clinical significancy with clinical relevency.

It would be better to delete references from conclusion section.

Author Response

1-) We agree with the reviewer#2. As the reviewer pointed out we changed "incidence" as "prevalence".

2-) Since this is a retrospective study, we collected all data of the Cypriot patients with field of views which demonstrated all maxilla (from the Apertura piriformis to alveolar crest). But for the sake of clarity we added the most common indications of our data to the materials-methods section.

3-) As the reviewer#3 suggested we replaced "clinical significancy" with "clinical relevency".

4-) We rewrote the end of the discussion and section and the beginning of the conclusion section in order to eliminate the references at the conclusion section as the reviewer#3 suggested.

Reviewer 3 Report

This is a retrospective CBCT study investigating the incidence, radiographic features, and clinical significance of accessory canals of the canalis sinuosus. The authors said they followed the EQUATOR guidelines in sample size calculation, meaningful difference between groups, sample preparation and handling, allocation sequence, randomization, and blinding statistical analysis. Two minor concerns have arisen:

1. The title should include the nationality of the patients (Cypriot population).

2. A table with the values of the parameters tested and their statistical significance should be added.  

Author Response

1-) As the reviewer#3 suggested we added "Cypriot Population" to the title.

2-) We already uploaded a table to the manuscript, however in order to make it clearer we revised our table.

3-) As reviewers suggested we revised our manuscript and corrected some typos.

Round 2

Reviewer 1 Report

I recommend accept for publication